# Predicting corporate management performance using AI: Incorporating CEO strategy insights from sustainable management reports

Xiao Wang[1], Feng Sun[2], Yong Ki Kim[3], Hyungjoon Kim[4], WonHo Song[4], Yubing Wei[5]*

1 School of Business Administration, Binzhou Polytechnic, Binzhou, Shandong, China, 2 School of Humanities, Binzhou Polytechnic, Binzhou, Shandong, China, 3 Department of Accounting and Taxation, Semyung University, Semyeong-ro, Jecheonsi, Chungcheongbukdo, Republic of Korea, 4 Department of Computer Engineering, Changwon National University, Changwonsi, Gyeongsangnamdo, Republic of Korea, 5 Department of Business Administration, Semyung University, Jecheonsi, Chungcheongbukdo, Korea

* weiwei992@naver.com

## Abstract

This study proposes an AI-based model to predict corporate management performance by combining financial data with strategic information extracted from CEO messages in sustainability reports. Using a dataset of 1,271 listed companies on Korea's KOSPI and KOSDAQ markets (2016–2023), we applied eight machine learning and deep learning classifiers: KNN, SVM, GBM, CatBoost, GAN, RNN, LSTM, and Transformer. Financial variables were selected based on prior accounting research, while strategic variables were derived via text mining of CEO messages and categorized using the Sustainable Balanced Scorecard (SBSC) framework. Results show that models incorporating both financial and strategy-based variables outperformed those using financial data alone. Notably, the Transformer model achieved the highest predictive accuracy, followed by LSTM and RNN. These findings provide actionable insights for investors and corporate stakeholders while advancing interdisciplinary research between accounting and AI. Under 5-fold cross-validation, the best-performing hybrid model (Transformer with SBSC features) achieved Accuracy=0.8467, AUC=0.8481, and F1=0.8572, and adding SBSC strategy indicators improved mean performance across models (ΔAccuracy=+0.0121; ΔAUC=+0.0092; ΔF1=+0.0119).

## Introduction

In recent years, environmental, social, and governance (ESG) considerations have emerged as a central focus of global corporate management [1]. Consequently, firms are increasingly formulating their management strategies in alignment with ESG principles [2]. Amid these evolving managerial paradigms, developing accurate predictive models for corporate performance has become a pressing academic and practical challenge.

**Data availability statement:** The data used in this study are available from the following sources: 1. Sustainability management reports provided by the ESG Portal Site (https://esg.krx.co.kr) 2. Financial and corporate information from NICE Credit Rating Information's Value Search (http://www.nicerating.com) In addition, the curated dataset constructed for this study, which was used in the experiments, is publicly available on Zenodo at https://zenodo.org/records/16865730.

**Funding:** The author(s) received no specific funding for this work.

**Competing interests:** The authors have declared that no competing interests exist.

While traditional approaches to performance forecasting have relied heavily on financial indicators [3,4], such data alone cannot fully capture a firm's strategic orientation or sustainability posture. Corporate reports often contain non-financial disclosures that provide meaningful insights into strategic direction. In particular, sustainability management reports have gained prominence as critical sources of ESG-related information. Notably, the CEO's message within these reports often articulates a company's long-term vision, goals, and strategic intent in a concise and representative manner [5–8].

Recent research suggests that ESG-related non-financial disclosures and corporate narratives contain forward-looking information that can complement traditional accounting signals in explaining or forecasting firm outcomes (ESG performance and financial outcomes [1,2]; sustainability reporting as a strategic communication channel [6]; CEO vision articulation and firm performance [7]; NLP-based analyses of CEO messages and sustainability reports [5,8,9]). Yet the evidence base remains fragmented: many studies focus on correlational links or descriptive insights, and fewer test whether strategy cues from sustainability-report CEO messages provide incremental predictive value beyond established financial predictors under modern machine/deep-learning benchmarks [3,4].

Accordingly, we highlight three gaps that motivate this study: (i) limited large-sample evidence on the incremental predictive value of CEO-message strategy narratives in sustainability reports beyond conventional accounting ratios; (ii) insufficient use of transparent, theory-grounded mappings such as SBSC to convert narrative content into interpretable strategic features; and (iii) scarce benchmarking of such hybrid text–tabular information across diverse ML/DL architectures. This study fills these gaps by mapping CEO-message keywords to SBSC categories to construct interpretable strategy indicators and by evaluating their added value across eight classifiers—including Transformer/LSTM—using 5-fold cross-validation on Korean listed firms (2016–2023).

However, despite the rapid expansion of ESG disclosure and the growing reliance of investors and regulators on sustainability information, there is still limited evidence on whether the strategic narratives embedded in sustainability reports can systematically improve firm-performance prediction beyond conventional accounting ratios [1,2,3,4]. Prior forecasting studies typically treat non-financial text as anecdotal or rely on less interpretable representations, leaving a methodological gap for a transparent, theory-grounded translation of CEO discourse into actionable strategic features. Addressing this gap is important because misclassifying future management performance can lead to inefficient capital allocation and delayed risk responses, especially in ESG-sensitive markets. Therefore, a hybrid and interpretable AI framework that integrates validated financial variables with SBSC-mapped CEO strategy signals is needed.

Unlike prior studies that utilize financial statements or MD&A sections to infer corporate strategies, this study leverages the CEO's message from sustainability management reports as the primary source for extracting strategic insights. This approach reflects the growing importance of ESG in corporate governance and

strategy formulation. By applying text mining techniques to these messages and classifying the extracted content using the Sustainable Balanced Scorecard (SBSC) framework, this study integrates qualitative strategic orientation into a machine learning-based corporate performance prediction model.

Furthermore, this research addresses the limitations of prior models that neglected the role of strategic, non-financial data in predictive accuracy. By quantifying the management strategy embedded in ESG disclosures, this study not only augments existing financial models but also introduces a hybrid approach that aligns with emerging sustainability paradigms. The proposed model is expected to offer enhanced predictive validity and provide valuable decision-making insights for corporate managers, investors, and stakeholders concerned with long-term performance and sustainable growth [10].

Predictive technology has advanced significantly due to the rapid development of artificial intelligence-based machine learning and deep learning technologies [11]. These technologies are particularly effective in analyzing diverse and complex datasets. In this context, should text mining techniques be applied to refine the company's financial data and the management strategy information from the CEO's message in the non-financial sustainability report, the model's predictive performance would be expected to improve significantly. Incorporating such refined information into the predictive model may further enhance its accuracy and reliability, as recent studies suggest that integrating structured and unstructured data contributes meaningfully to performance improvements.

In other words, this paper aims to develop an AI-based prediction model that can more accurately predict a company's management performance by reflecting the information contained in the CEO message of the sustainability management report. The management performance prediction model that reflects management strategy information, including corporate financial data and ESG activities, is expected to perform better than existing models. This model is systematically and logically designed using financial variables that have already been validated in prior accounting research.

In summary, the aim of this study was to develop an AI-based predictive model which would predict the performance management of a company with accuracy by analyzing the information normally contained within the CEO's report on sustainability management. The proposed model, which integrates strategic management information—particularly financial data and ESG activities—is anticipated to outperform traditional models in forecasting managerial performance. Furthermore, this study's findings are expected to inspire new academic research in predictive analytics and bridge interdisciplinary gaps between business administration and computer engineering. The results also offer practical value by supporting more informed decision-making for various corporate stakeholders.

This study differs from prior research in several meaningful ways, both methodologically and conceptually. First, while earlier studies often utilized broad financial datasets without a strong foundation in accounting theory, this study constructs a predictive model grounded in validated financial variables derived from accounting literature, focusing on those with a direct link to management performance.

Second, this research uniquely incorporates strategic, non-financial insights by extracting and analyzing CEO messages from sustainability management reports using text mining techniques—a novel approach not previously applied in this context. While CEO messages are often seen as symbolic or aspirational, this study demonstrates that when systematically processed, they contain strategic signals that can meaningfully contribute to performance prediction. This approach aligns with current trends emphasizing the role of ESG and sustainability discourse in corporate decision-making.

Third, by applying eight machine learning and deep learning classification models—including Transformer and LSTM architectures—the study provides a robust comparative framework. Results show that models incorporating CEO message data outperform those relying solely on financial variables, indicating that strategic narratives provide additional explanatory power.

From a practical standpoint, the methodology can be applied in investment analysis, strategic forecasting, and stakeholder decision-making, especially in ESG-focused evaluations. Academically, this research contributes to the literature by

integrating textual sustainability disclosures with quantitative performance modeling, bridging gaps between accounting, strategic management, and AI-driven analytics.

This paper has academic value because it distinguishes itself from previous studies. Its results can present data necessary for establishing a management strategy that considers a company's ESG activities and provides meaningful information for sustainable growth.

## Research design

### Data

The publicly available sustainability management report provides essential details about a company's ESG activities, vision, and strategic objectives [6]. Of particular interest is the CEO's message, which often implicitly reveals the company's long-term strategic direction [5].

In this study, text data from sustainability management reports was extracted using text mining techniques and further processed with Natural Language Processing (NLP) methods [9]. Specifically, the NLP toolkit used was NLTK (Natural Language Toolkit), which supported preprocessing tasks such as text normalization, tokenization, and stop word removal [12,13]. Text normalization removed punctuation and special characters, while stop word removal filtered out irrelevant terms. Tokenization segmented the text into meaningful units.

To numerically represent the processed textual data, the Bag of Words (BoW) model was employed. BoW converts text into numerical vectors by mapping word frequencies or presence in a document based on a vocabulary of unique words, ignoring word order and grammar, thus simplifying the text into a form suitable for statistical analysis [14]. For implementation, Scikit-learn's CountVectorizer and TfidfVectorizer were used to extract frequency-based features and compute TF-IDF (Term Frequency-Inverse Document Frequency) values [15].

We selected a transparent and widely used NLP toolchain (NLTK plus scikit-learn vectorizers) because our objective is to extract interpretable, SBSC-mappable keywords from relatively short CEO messages and to ensure full reproducibility. A BoW/TF-IDF representation is well suited to keyword salience extraction and avoids the data-hungriness of embedding-heavy approaches when the sample size is moderate. The SBSC framework provides a theory-grounded mapping from textual cues to strategic dimensions, yielding low-dimensional dummy indicators that can be cleanly integrated with accounting variables. This combination balances interpretability, methodological rigor, and feasibility for firm-level sustainability disclosures,

Alternative text-analytics options for CEO messages include topic modeling, sentiment scoring, or dense embedding-based representations. However, CEO messages in sustainability reports are relatively short and highly domain-specific; embedding-heavy pipelines often require larger corpora and typically yield less transparent features for strategy interpretation. In contrast, a TF-IDF keyword approach combined with SBSC mapping provides a theory-grounded and auditable bridge from narrative cues to interpretable strategic indicators that can be directly integrated with accounting variables. We also considered using the full sustainability report text (or other narrative sections such as MD&A), but we focus on the CEO message because it offers a concise, comparable statement of strategic intent across firms and years, reducing heterogeneity in document length and structure.

Keywords with TF-IDF values of 1.5 or higher were extracted and categorized under the SBSC (Sustainability Balanced Scorecard) framework. This framework classifies the strategic orientation into five categories: financial, customer, internal process, learning and growth, and sustainability [16,17]. The dominant keyword category in each CEO's message determined the company's strategic emphasis. For example, if a CEO's message included five keywords aligned with the financial perspective, three with customer, and two with internal process, the strategy was classified as finance-focused. A binary indicator of 1 was assigned to the financial variable, and 0 to the others.

To justify the TF-IDF cutoff, we conducted a sensitivity check across several candidate thresholds and reviewed the resulting keyword sets in terms of (i) keyword sparsity per CEO message, (ii) the extent to which generic, broadly used

terms remained, and (iii) the stability of the SBSC category assignment. Thresholds below 1.5 produced excessively large and less discriminative keyword lists, whereas thresholds above 1.5 often yielded overly sparse sets that could destabilize SBSC coding. Accordingly, 1.5 was selected as a practical balance that retains salient terms while filtering out low-information tokens; the main findings were qualitatively robust to nearby thresholds.

The management strategy data was sourced from sustainability management reports available on the ESG Portal Site (https://esg.krx.co.kr), with financial data obtained Value Search platform (https://www.nicevse.com/vse/main.html). The analysis covered data from 2016 to 2023 for companies listed on KOSPI and KOSDAQ.

These data sources are publicly accessible, and the organized dataset used in this study is openly available at Zenodo (DOI: https://doi.org/10.5281/zenodo.16865730).

From an initial pool of 16,343 companies listed on KOSPI and KOSDAQ between 2016 and 2023, a series of exclusion criteria were applied to ensure the integrity and consistency of the analytical dataset. First, firms in the financial industry were excluded due to their distinct financial structures, regulatory requirements, and reporting standards, which could introduce sector-specific biases and reduce the homogeneity of the sample. Second, 385 companies that did not close their fiscal year in December were removed to maintain consistency in financial reporting periods, thereby enhancing the comparability of financial data.

Third, 993 companies without available financial data were excluded, as financial variables constitute a foundational input for the predictive model developed in this study. Fourth, 35 firms exhibiting capital erosion due to sustained losses were excluded to prevent extreme or distressed cases from distorting the analytical outcomes. This step ensured that the analysis focused on firms operating under standard financial conditions.

Fifth, 8,455 companies lacking ESG ratings from the Korea ESG Standards Institute were excluded, since ESG ratings serve as essential non-financial inputs for this model, reflecting companies' sustainability performance. Lastly, 5,604 companies that did not disclose a CEO message within their sustainability reports were omitted. As this study derives strategic management insights primarily from CEO messages, the absence of such content rendered these firms incompatible with the study's analytical framework.

These exclusion criteria collectively enhanced the methodological rigor of the study by producing a clean, homogenous dataset aligned with the model's design assumptions and analytical goals. Table 1 below shows the sample information utilized in this empirical analysis. Table 1 reports the stepwise construction of the empirical sample of Korean listed firms (KOSPI/KOSDAQ) over 2016–2023; negative values indicate exclusions at each filtering step (e.g., non-December fiscal year-end, missing financial data, missing ESG ratings, or missing CEO messages).

Table 2 summarizes the variables used in this paper's research model. ROE measures corporate management performance, and the research focuses on developing machine learning and deep learning models to predict the firm performance. The meaning of the most frequently used keywords in this study is that there are relatively many keywords with high TF-IDF values. The management strategy variables used in this study are designed to capture specific aspects of a

**Table 1. Corporate sample selection.**

| Sample selection | Sample number |
| --- | --- |
| Listed companies excluding the financial industry from 2016 to 2023 | 16,343 |
| Excluding companies for December settlement | −385 |
| Excluding enterprises that erode capital | −35 |
| Exclude companies without corporate financial data | −993 |
| Excluding companies that do not disclose ESG ratings by the Korea ESG Standards Institute | −8,455 |
| Excluding companies that have not disclosed sustainability management reports | −5,604 |
| Final sample from 2016 to 2023 | 871 |

**Table 2. Variables definition.**

| Variables | Definition |
|---|---|
| ROE | Return on equity = Net income/ Total capital |
| Financial | Dummy variable, if the most keywords for finance in the Sustainable Management Report's CEO message is 1, otherwise 0 |
| Customer | Dummy variable, if the most keywords for customers in the Sustainable Management Report's CEO message is 1, otherwise 0 |
| Internal process | Dummy variable, if the most keywords for Internal process in the Sustainable Management Report's CEO message is 1, otherwise 0 |
| Learning & Growth | Dummy variable, if the most keywords for Learning & Growth in the Sustainable Management Report's CEO message is 1, otherwise 0 |
| Sustainability | Dummy variable, if the most keywords for sustainability in the Sustainable Management Report's CEO message is 1, otherwise 0 |
| SIZE | Natural logarithmic value of total assets |
| Leverage | Debt ratio = Total liabilities/ Total assets |
| Current | Current Ratio = Current Liabilities/ Current Assets |
| Growth | The rate of increase/decrease in sales |
| Foreigner | Foreign shareholders' equity ratio |
| Largest | The largest shareholder's equity ratio |
| BIG4 | If the accounting firm that conducted the external audit is a major accounting firm corresponding to BIG4, a dummy variable that means 1 or 0 |
| ESG | Scores based on the ESG rating of companies disclosed by the Korea ESG Standards Institute (A+=6, A=5, B+=4, B=3, C=2, D=1) |

company's management strategy delivered through the CEO's message in the sustainability report concerning previous studies [5]. Table 2 defines the dependent variable (ROE), five SBSC-based management-strategy indicators derived from CEO-message TF-IDF keywords (dummy variables), and the accounting/ESG control variables used as model inputs; it also clarifies each variable's measurement and coding.

The Finance, Customer, Internal Process, Learning & Growth, and Sustainability variables are all dummy variables. These variables are assigned a value of 1 if the most frequently repeated keywords in the CEO's message of the Sustainable Management Report are related to the respective categories; otherwise, the value is 0. Specifically.

• Finance is 1 if the dominant keywords are related to finance, and 0 otherwise.

• Customer is 1 if the dominant keywords relate to customers, and 0 otherwise.

• Internal Process is 1 if the dominant keywords pertain to internal processes, and 0 otherwise.

• Learning & Growth is 1 if the dominant keywords are linked to employee education and growth, and 0 otherwise.

• Sustainability is 1 if the most frequent keywords are related to environmental issues, donations, or sustainable growth, and 0 otherwise.

Table 3 provides the descriptive statistics for the variables used in the empirical analysis. The mean value of ROE, which measures corporate management performance, is 0.065, with a standard deviation of 0.136. The Financial variable, indicating the company's financial focus in its management strategy, has a mean of 0.194 and a standard deviation of 0.396. The Customer variable, which emphasizes customer-centric management strategies, shows a mean value of 0.233 with a standard deviation of 0.423. The Internal variable, which focuses on internal processes within the company's management strategy, has an average value of 0.219 and a standard deviation of 0.414. The Learning & Growth variable,

**Table 3. Descriptive statistics.**

| Variables | N | Mean | Std | Min | Q1 | Median | Q3 | Max |
|---|---|---|---|---|---|---|---|---|
| ROE | 871 | 0.065 | 0.136 | −0.831 | 0.018 | 0.061 | 0.117 | 0.561 |
| Finance | 871 | 0.194 | 0.396 | 0.000 | 0.000 | 0.000 | 0.000 | 1.000 |
| Customer | 871 | 0.233 | 0.423 | 0.000 | 0.000 | 0.000 | 0.000 | 1.000 |
| Internal | 871 | 0.219 | 0.414 | 0.000 | 0.000 | 0.000 | 0.000 | 1.000 |
| Learning & Growth | 871 | 0.233 | 0.423 | 0.000 | 0.000 | 0.000 | 0.000 | 1.000 |
| Sustainability | 871 | 0.215 | 0.411 | 0.000 | 0.000 | 0.000 | 0.000 | 1.000 |
| SIZE | 871 | 28.876 | 1.335 | 25.016 | 27.961 | 28.948 | 30.056 | 30.604 |
| Leverage | 871 | 0.432 | 0.199 | 0.027 | 0.266 | 0.445 | 0.579 | 0.875 |
| Current | 871 | 2.056 | 3.637 | 0.195 | 0.854 | 1.247 | 1.965 | 32.472 |
| Growth | 871 | 0.078 | 0.328 | −0.701 | −0.050 | 0.041 | 0.154 | 2.111 |
| Foreigner | 871 | 0.199 | 0.161 | 0.002 | 0.077 | 0.152 | 0.271 | 1.000 |
| Largest | 871 | 0.408 | 0.162 | 0.064 | 0.300 | 0.387 | 0.511 | 0.917 |
| BIG4 | 871 | 0.897 | 0.305 | 0.000 | 1.000 | 1.000 | 1.000 | 1.000 |
| ESG | 871 | 4.423 | 0.954 | 1.000 | 4.000 | 5.000 | 5.000 | 6.000 |

reflecting the company's focus on learning and growth strategies, averages 0.233 with a standard deviation of 0.423. Lastly, the Sustainability variable, representing the company's emphasis on sustainability initiatives such as social responsibility and environmental protection, has a mean value of 0.215 and a standard deviation of 0.411. These statistics summarize the central tendencies and variability of the key variables used in the analysis. Table 3 summarizes the distribution of all modeling variables (e.g., mean, dispersion, and quartiles) for the final analytical sample; for binary SBSC indicators, the mean can be interpreted as the proportion of firm-years classified into each strategy category.

## Dataset adequacy for deep learning

To address concerns regarding dataset adequacy for deep learning application, this study employed several methodological safeguards. Although the dataset size is relatively modest compared to typical deep learning benchmarks, it comprises systematically curated financial and non-financial data extracted through validated text mining procedures. To mitigate potential overfitting, regularization, dropout, and early stopping techniques were applied during training. In addition, all models were subjected to cross-validation, and their performance was benchmarked against traditional machine learning classifiers. Notably, Transformer and LSTM models consistently outperformed other models, indicating that the integration of strategic textual features enhances predictive accuracy even within a limited dataset context. These measures collectively enhance the robustness and reliability of the model results.

The dataset is appropriate for this study because it directly matches the research question—predicting firm management performance using a hybrid of validated accounting covariates and CEO-message strategy cues. Our final analytical sample consists of 871 non-financial firms listed on Korea's KOSPI and KOSDAQ markets over 2016–2023 (Table 1), a period in which sustainability reporting became increasingly institutionalized. This coverage provides meaningful cross-sectional and sectoral heterogeneity while maintaining comparability through consistent fiscal-year alignment (December year-end) and standardized exclusion rules (e.g., removing firms with missing key inputs or severe distress).

In addition, the dataset integrates three complementary, decision-relevant information sources that are widely used by market participants: (i) audited firm financial statements obtained from a commercial database, (ii) ESG ratings released by the Korea ESG Standards Institute, and (iii) CEO messages from publicly available sustainability management reports on the KRX ESG portal. This triangulation increases construct validity by combining outcome-relevant financial fundamentals with independently assessed ESG performance and forward-looking managerial narratives. From a modeling

 

perspective, the text component is converted into a low-dimensional, theory-grounded representation via SBSC mapping (five dummy indicators), which reduces noise and limits the risk of high-dimensional sparsity. Moreover, the binary outcome definition (ROE above vs. below the sample mean) yields a near-balanced class split, and generalization is assessed using 5-fold cross-validation together with regularization, dropout, and early stopping, supporting the adequacy of the dataset for benchmarking both ML and DL models.

**Prediction models**

This study draws on previous research on corporate management performance to construct a model for predicting such performance. The key variables in the standard model are outlined below,

SIZE variable represents a company's size and is calculated by taking the natural logarithm of its total assets. Due to their scale, larger firms typically demonstrate more stable management performance [18]. Leverage variable indicates the ratio of a company's debt to its total assets. Generally, a higher debt ratio is associated with a weaker financial position and more volatile management performance [19]. Current represents a firm's liquidity ratio, and this variable indicates financial stability. A company's liquidity can significantly impact its management strategy and overall performance [20]. Growth refers to the company's sales growth rate. Higher sales growth is often linked to improved management performance, leading to greater profitability [21]. Foreigner variable captures the proportion of shares held by foreign shareholders. A higher percentage of foreign ownership often leads to better monitoring, contributing to a more transparent governance structure that can enhance management performance [22]. Largest variable represents the ownership percentage of the largest shareholder. While higher ownership by the largest shareholder can lead to more effective corporate management, some studies suggest that it can also result in poorer corporate governance, negatively impacting management [23, 24].BIG4 is a dummy variable indicating whether a major accounting firm audited the company. Audits by prominent firms are often linked to larger company size and improved management performance [25]. ESG variable represents the company's ESG (Environmental, Social, and Governance) rating, as disclosed by the Korea ESG Standards Institute [26]. This study quantifies ESG grades on a scale from A+ (6) to D (1). A higher ESG rating generally correlates with better management performance, as it enhances the company's image and, over the long term, is likely to sustain or improve management outcomes.

This research compares the predictive performance of a standard model with an enhanced model that incorporates management strategy information derived from sustainability management reports. The two corporate management performance prediction models are expressed as functions in (1) and (2).

[The standard model for forecasting firm management performance]

$$F(SIZE,\ Leverage,\ Current,\ Growth,\ Foreigner,\ Largest,\ BIG4,\ ESG) = ROE \tag{1}$$

[The model that includes management strategy information to the standard model for forecasting firm management performance]

$$F\left(\begin{array}{c}SIZE,\ Leverage,\ Current,\ Growth,\ Foreigner,\ Largest,\ BIG4,\ ESG,\ Finance,\\ Customer,\ Internal,\ Learning\ Growth,\ Sustainability\end{array}\right) = ROE \tag{2}$$

The following Fig 1 is a schematic diagram of how to compare and analyze the performance of the corporate management performance prediction model using the machine learning and deep learning classification method in this paper. This study uses eight machine learning and deep learning classifiers: K-Nearest Neighbors (KNN), Support Vector Machine (SVM), Gradient Boosting Machine (GBM), CatBoost, Generative Adversarial Networks (GAN), Recurrent Neural Networks (RNN), Long Short-Term Memory Networks (LSTM), Transformer.

Fig 1 summarizes the end-to-end workflow: extracting CEO-message text from sustainability reports, computing TF-IDF keywords and mapping them to SBSC categories to form five strategy indicators, combining these indicators with

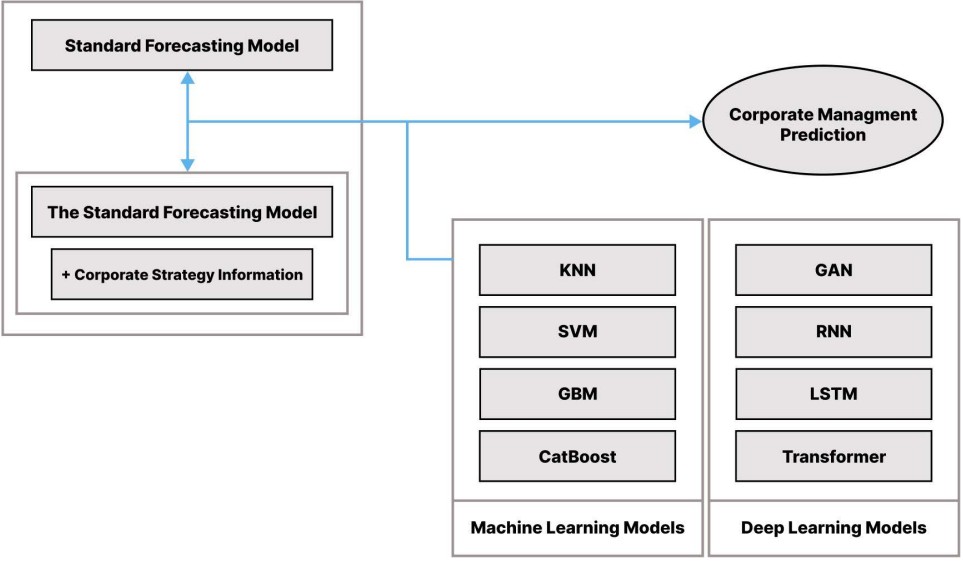

**Fig 1. Research model.**

accounting and ESG variables, and benchmarking eight classifiers under 5-fold cross-validation to compare the standard and hybrid models. The standard model uses validated accounting predictors (SIZE, Leverage, Current, Growth, Foreigner, Largest, BIG4) and the ESG rating, whereas the hybrid model additionally includes five SBSC-based strategy dummy indicators (Finance, Customer, Internal Process, Learning & Growth, Sustainability) derived from CEO-message TF-IDF keywords. The prediction target is a binary label of management performance (ROE above vs. below the sample mean), and models are evaluated via 5-fold cross-validation with multiple metrics (Accuracy, Precision, Recall, AUC, and F1, Tables 5–9).

## Classification method

This study adopts a binary classification approach to predict corporate management performance. Specifically, firms with a Return on Equity (ROE) above the sample mean are labeled as "1" (high performance), while those below the mean are labeled as "0" (low performance). This methodological choice is justified on both theoretical and practical grounds.

First, binary classification reflects real-world corporate evaluation and decision-making frameworks. In practice and academic contexts, dichotomous classifications such as "high vs. low," "strong vs. weak," or "above vs. below threshold" are commonly used in credit ratings, risk assessments, and strategic benchmarking. Such classification enhances both interpretability and practical applicability of the analysis.

Second, the non-financial strategic information extracted from CEO messages via text mining techniques is inherently qualitative and symbolic rather than strictly quantitative. As such, this type of strategic signal is more appropriately analyzed through a classification model that determines the presence or absence of impact on performance, rather than a regression model that assumes continuous effects.

Lastly, deep learning classifiers such as LSTM and Transformer exhibit superior predictive performance in environments combining structured financial data with unstructured text data. Using a binary outcome variable allows these models to maximize learning efficiency and prediction accuracy, making them well-suited for this task.

In sum, the binary classification approach employed in this study is both theoretically sound and practically relevant, offering a robust methodology for improving prediction performance and enhancing the applicability of the model to real-world decision-making contexts.

Cross-validation is a robust method for evaluating model performance, effectively mitigating the risks of overfitting and underfitting. This technique systematically partitions the dataset into multiple subsets, each sequentially employed for training and validation, thereby comprehensively assessing the model's generalization capabilities. Additionally, cross-validation facilitates the evaluation of the model across diverse data patterns instead of relying solely on a single train-test split. This approach establishes an essential baseline for the early identification of potential overfitting or underfitting and supports the optimization of the model. In this study, the K-fold cross-validation method, widely recognized for its reliability, was utilized with K set to 5. This method divided the dataset into five folds, each serving as the validation set once, while the remaining folds were used for training.

This paper investigates classification techniques to develop machine learning and deep learning models for predicting firm management performance. In machine and deep learning, classification involves assigning input data to predefined categories or classes [27,28]. Typically, models employ softmax functions in the final output layer to produce probability scores for each class, with the highest probability being selected for the prediction [29].

Machine learning and deep learning models often rely on a cross-entropy loss function to solve classification tasks, quantifying the difference between the predicted class probabilities and the actual class labels [30]. The model parameters are adjusted to minimize this difference, enhancing the model's accuracy [31]. The performance of these models is evaluated using key metrics such as accuracy, precision, recall, AUC (Area Under the ROC Curve), and F1 score [32].

Accuracy indicates the frequency with which the model's predictions align with the actual class labels, serving as a general measure of the model's effective-ness [33]. Precision refers to the ratio of true positive predictions to all positive predictions, thereby assessing the model's accuracy in identifying positive instances [34]. Recall measures the proportion of true positive cases correctly identified by the model, indicating its ability to capture all relevant instances [35]. AUC evaluates binary classification models by assessing their ability to distinguish between positive and negative classes through the ROC (Receiver Operating Characteristic) curve [36]. This curve plots the True Positive Rate (TPR) against the False Positive Rate (FPR) at various classification thresholds [37]. AUC values range from 0 to 1, with higher values indicating better model performance [36]. The F1 score is the harmonic mean of precision and recall, providing a balanced assessment of the model's classification performance, particularly in cases with imbalanced data [38]. The F1 score also ranges from 0 to 1, with higher values indicating superior classification ability [39].

This study highlights the significance of these metrics in evaluating the effectiveness of machine learning models for predicting management performance. It offers insights into selecting and optimizing classification techniques to enhance predictive accuracy.

## Classifier

Choice of methods relative to alternatives. Our objective is not only to maximize predictive accuracy, but also to assess whether SBSC-based strategy cues add incremental, interpretable signal beyond established financial predictors. For this reason, we adopt a hybrid tabular-plus-strategy design rather than an end-to-end, high-dimensional text model that could obscure the contribution of specific strategic dimensions. For the tabular component, gradient-boosting ensembles (GBM/CatBoost) are strong, widely used baselines for structured corporate data, while SVM provides a representative margin-based comparator; additional options such as logistic regression or random forests would be largely redundant given these baselines. For the deep-learning component, RNN/LSTM and Transformer models are included because they can capture nonlinear interactions and complex decision boundaries, allowing us to benchmark whether higher-capacity architectures yield consistent gains in this hybrid setting. Finally, 5-fold cross-validation and multiple

complementary metrics (Accuracy, AUC, and F1) are used instead of a single split to provide stable out-of-sample estimates and to ensure that conclusions are robust to threshold choice and class balance,

Justification for the different predictive modeling tools. We benchmarked eight classifiers spanning complementary inductive biases: distance-based (KNN), margin-based (SVM), tree-ensemble boosting for tabular data (GBM and CatBoost), adversarial/generative modeling (GAN), sequence models (RNN and LSTM), and attention-based architectures (Transformer). This breadth reduces the risk that conclusions are driven by a single modeling assumption and is appropriate for our hybrid feature set combining structured financial covariates with SBSC-based strategy indicators. We used 5-fold cross-validation to obtain stable out-of-sample estimates given the sample size and to diagnose overfitting/underfitting consistently across model families, and we report both threshold-dependent and threshold-independent metrics (Accuracy, Precision, Recall, AUC, and F1) to reflect potential class imbalance and differing decision-use cases.

Cross-validation is a reliable method for assessing a model's performance while guarding against overfitting and underfitting [40]. This technique divides the dataset into multiple segments each used alternately for training and validation. This process helps to obtain a robust evaluation of the model's ability to generalize to unseen data [41].

Furthermore, cross-validation enables the model to be tested on various data patterns rather than restricted to a single train-test split [42]. This approach is crucial for detecting overfitting or underfitting and helps fine-tune the model for better performance.

This study employed the widely used K-fold cross-validation method, with K set to 5. This approach splits the dataset into five folds, each used as a validation set once, and the remaining folds serve as the training set. This ensures that every part of the dataset is used for training and validation, comprehensively assessing the model's performance.

K-Nearest Neighbors (KNN) is a straightforward, non-parametric algorithm for classification and regression [43]. The fundamental principle is classifying a data point by considering the majority class among its k nearest neighbors within the feature space [44]. In regression tasks, it averages the values of these k neighbors [43]. While KNN is simple and intuitive, its computational cost can become significant as the dataset size increases since it requires calculating distances to every other point [45].

Support Vector Machine (SVM) is a supervised learning algorithm primarily applied to classification problems [46]. It identifies the hyperplane that most effectively separates the data into distinct classes [47]. The optimal hyperplane maximizes the margin between the classes [46]. SVMs perform well in high-dimensional spaces, making them particularly effective when the number of dimensions exceeds the number of samples [48]. However, they may struggle with noisy data and are sensitive to the choice of kernel [47].

Gradient Boosting Machine (GBM) is an ensemble technique that constructs models sequentially, with each new model correcting the errors of its predecessor [49]. The "boosting" refers to combining models to enhance accuracy [50]. GBM is widely used for tasks involving structured data, such as classification, regression, and ranking [51]. Despite its ability to produce strong predictive models, GBM is prone to overfitting and requires careful tuning of hyperparameters [49,50].

CatBoost is a gradient-boosting algorithm specifically tailored to handle categorical data [52]. It processes categorical features directly without converting them into numerical values, such as through one-hot encoding [53]. Compared to other boosting algorithms, CatBoost stands out for its high performance and reduced need for extensive hyperparameter tuning [52]. It is particularly favored in fields like finance and e-commerce, where categorical data is prevalent [54].

Generative Adversarial Network (GAN) involves two neural networks, a generator and a discriminator, that engage in a competitive learning process [55]. The generator produces synthetic data samples, while the discriminator distinguishes between real and synthetic data [56]. This adversarial approach allows GANs to generate highly realistic data, including images and videos [57]. GANs are extensively used in image generation, style transfer, and data augmentation, though they can be challenging to train and require careful tuning of hyperparameters [55,57].

Motivation for using GAN in this study. Although the present task is predictive classification (high vs. low management performance) rather than pure data synthesis, we included a GAN-based model as an additional deep-learning baseline for three reasons. First, tabular corporate/financial datasets are often moderate in size and may exhibit class imbalance; adversarial learning and GAN-based synthetic data generation have been shown to improve downstream classification by enriching minority-class samples and stabilizing decision boundaries. Second, modern conditional GAN variants (e.g., CTGAN) are designed to model mixed-type tabular distributions (continuous and categorical), making them suitable for accounting and non-financial variables in firm-level prediction settings. Third, prior studies in finance and risk analytics report that GAN-driven synthetic data and augmentation can enhance predictive robustness. Accordingly, we evaluated a GAN model alongside other deep learners under the same cross-validation protocol to benchmark its utility in our setting [58–60].

Recurrent Neural Network (RNN) is designed to process sequential data [61]. Unlike feedforward networks, RNNs have cyclical connections that allow them to retain the memory of previous inputs, making them well-suited for tasks like time series prediction, natural language processing, and speech recognition [62]. However, RNNs can encounter issues such as vanishing and exploding gradients, which can complicate the training process [63].

Long-short-term memory (LSTM) networks are a specialized type of RNN engineered to handle long-term dependencies in sequence data [64]. LSTM uses gating mechanisms to manage the flow of information, allowing them to preserve important information over extended sequences and mitigate the vanishing gradient problem [65]. It is widely used in text generation, machine translation, and forecasting time series [66].

Transformer is a neural network architecture that leverages self-attention mechanisms to process sequential data [67]. Unlike RNN, the transformer does not rely on sequential data processing; instead, it enables the model to evaluate the entire sequence simultaneously by computing attention scores [68]. This approach makes transformers highly efficient and effective for language translation, text summarization, and speech recognition tasks [69]. Transformers form the basis of many cutting-edge models in natural language processing, such as BERT and GPT [67].

Table 4 provides a concise, self-contained description of the eight benchmark classifiers used in this study (KNN, SVM, GBM, CatBoost, GAN, RNN, LSTM, and Transformer), highlighting their core learning assumptions to support methodological comparability.

**Table 4. The core concept of the classifiers.**

| Classifier | Core Concept |
|---|---|
| K-Nearest Neighbors (KNN) | Classifies based on majority vote among k nearest neighbors; averages values for regression. |
| Support Vector Machine (SVM) | Finds optimal hyperplane that maximizes margin between classes; effective in high-dimensional space. |
| Gradient Boosting Machine (GBM) | Sequential ensemble method correcting errors of prior models; strong with structured data. |
| CatBoost | Handles categorical data natively; high performance with reduced need for hyperparameter tuning. |
| Generative Adversarial Network (GAN) | Uses generator and discriminator in adversarial training to produce realistic synthetic data. |
| Recurrent Neural Network (RNN) | Processes sequential data using cyclic connections; maintains memory of previous inputs. |
| Long Short-Term Memory (LSTM) | Specialized RNN that handles long-term dependencies using gating mechanisms. |
| Transformer | Uses self-attention to process entire sequences in parallel; basis of modern NLP models. |

Each model has distinct advantages and applications, making it suitable for various machine learning tasks depending on the specific problem requirements. Given the diverse characteristics, strengths, and limitations of these machine learning and deep learning classification algorithms, selecting and applying the appropriate algorithm based on the problem and dataset is crucial. This study uses eight machine learning classifiers to evaluate the performance of the firm's management performance prediction model, which incorporates management strategy information from business reports.

## Results

Table 5 below presents the accuracy results for various models that predict corporate management performance. The KNN standard model initially achieved an accuracy of 0.7704, which increased to 0.7821 after incorporating management strategy information derived from the CEO messages in the sustainability management report. The SVM standard model had an accuracy of 0.7836, which improved to 0.7985 when management strategy information from the SBSC framework was added. The accuracy of the GBM standard model was 0.8007, which rose to 0.8141 with the inclusion of additional management strategy data. The accuracy of the CatBoost standard model was 0.7928, which improved to 0.8069 after incorporating management strategy information. The GAN model had a baseline accuracy of 0.8169, which increased to 0.8255 with the addition of management strategy data. The RNN model initially showed an accuracy of 0.8218, which improved to 0.8390 when management strategy information was included. The LSTM model had a standard accuracy of 0.8315, which rose to 0.8485 after integrating the SBSC framework's management strategy information. Finally, the Transformer model initially had an accuracy of 0.8467, which improved to 0.8467 by including management strategy data. These results highlight the positive impact of incorporating management strategy information on the predictive accuracy of various machine learning and deep learning models. The values of Table 5 are mean accuracy across 5-fold cross-validation. The standard model uses accounting and ESG predictors, while the hybrid model additionally includes SBSC-based strategy indicators extracted from CEO messages; higher values indicate better classification performance.

Table 6 provides the precision scores for predicting corporate management performance across various models. The KNN standard model achieved a precision of 0.7623, which increased to 0.7717 after incorporating management strategy information from the CEO messages in the sustainability management report. The SVM standard model demonstrated a precision of 0.7734, which improved to 0.7870 by adding management strategy information based on the SBSC framework. The GBM standard model's precision in predicting corporate management performance was 0.7964. This increased to 0.8042 after integrating management strategy data. The CatBoost standard model had a precision score of 0.7891, which rose to 0.7854 when management strategy information was included. The GAN model recorded a precision of 0.8063 in predicting management performance, which improved to 0.8152 by adding management strategy information. The RNN standard model initially had a precision of 0.8114, which increased to 0.8287 when management strategy data was incorporated. The precision score for the LSTM standard model was 0.8274, which improved to 0.8390 after including management strategy information according to the SBSC framework. Finally, the Trans-former standard model achieved a precision of 0.8353, which increased to 0.8482 with the inclusion of management strategy information. These results demonstrate that integrating management strategy information generally enhances the precision of various models in predicting corporate management performance. The values of Table 6 are mean precision across 5-fold cross-validation. The

**Table 5. Firm management performance prediction accuracy results.**

| Model | Classifier | Accuracy | Classifier | Accuracy | Classifier | Accuracy | Classifier | Accuracy |
|---|---|---|---|---|---|---|---|---|
| **Standard model** | KNN | 0.7704 | SVM | 0.7836 | GBM | 0.8007 | CatBoost | 0.7928 |
| **Standard model+Management strategy information** | | 0.7821 | | 0.7985 | | 0.8141 | | 0.8069 |
| **Standard model** | GAN | 0.8169 | RNN | 0.8218 | LSTM | 0.8315 | Transformer | 0.8467 |
| **Standard model+Management strategy information** | | 0.8255 | | 0.8390 | | 0.8485 | | 0.8467 |

**Table 6. Firm management performance prediction precision results.**

| Model | Classifier | Precision | Classifier | Precision | Classifier | Precision | Classifier | Precision |
|---|---|---|---|---|---|---|---|---|
| **Standard model** | KNN | 0.7623 | SVM | 0.7734 | GBM | 0.7964 | CatBoost | 0.7891 |
| **Standard model + Management strategy information** | | 0.7717 | | 0.7870 | | 0.8042 | | 0.7854 |
| **Standard model** | GAN | 0.8063 | RNN | 0.8114 | LSTM | 0.8274 | Transformer | 0.8353 |
| **Standard model + Management strategy information** | | 0.8152 | | 0.8287 | | 0.8390 | | 0.8482 |

standard model uses accounting and ESG predictors, while the hybrid model additionally includes SBSC-based strategy indicators extracted from CEO messages; higher values indicate fewer false positives among predicted positives.

Table 7 displays various models' recall values for predicting corporate management performance. The KNN standard model achieved a recall of 0.7792, which improved to 0.7884 by including management strategy information extracted from the CEO messages in the sustainability management report. The SVM standard model had a recall of 0.7862, which increased to 0.7990 when management strategy information based on the SBSC framework was incorporated. The recall for the GBM standard model was 0.7999, which was enhanced to 0.8123 by adding management strategy data. The CatBoost standard model initially showed a recall of 0.7910, which increased to 0.8087 upon incorporating management strategy information. The GAN model had a baseline recall of 0.8132, which improved to 0.8269 after adding management strategy information. The RNN model started with a recall of 0.8224, which rose to 0.8387 with the inclusion of management strategy information. The recall for the LSTM model was 0.8369, which improved to 0.8474 when management strategy information from the SBSC framework was included. Lastly, the Transformer model achieved a recall of 0.8486, which improved to 0.8575 by incorporating management strategy information. These results indicate that adding management strategy information generally enhances recall across different models, improving their ability to correctly identify positive corporate management performance. The values of Table 7 are mean recall across 5-fold cross-validation. The standard model uses accounting and ESG predictors, while the hybrid model additionally includes SBSC-based strategy indicators extracted from CEO messages; higher values indicate fewer false negatives among actual positives.

Table 8 provides the AUC results for predicting corporate management performance across various models. The KNN standard model achieved an AUC of 0.7685, which improved to 0.7748 when management strategy information from the CEO messages in the sustainability management report was included. The SVM standard model demonstrated an AUC of

**Table 7. Firm management performance prediction recall results.**

| Model | Classifier | Recall | Classifier | Recall | Classifier | Recall | Classifier | Recall |
|---|---|---|---|---|---|---|---|---|
| **Standard model** | KNN | 0.7792 | SVM | 0.7862 | GBM | 0.7999 | CatBoost | 0.7910 |
| **Standard model + Management strategy information** | | 0.7884 | | 0.7990 | | 0.8123 | | 0.8087 |
| **Standard model** | GAN | 0.8132 | RNN | 0.8224 | LSTM | 0.8369 | Transformer | 0.8486 |
| **Standard model + Management strategy information** | | 0.8269 | | 0.8387 | | 0.8474 | | 0.8575 |

**Table 8. Firm management performance prediction AUC results.**

| Model | Classifier | AUC | Classifier | AUC | Classifier | AUC | Classifier | AUC |
|---|---|---|---|---|---|---|---|---|
| **Standard model** | KNN | 0.7685 | SVM | 0.7762 | GBM | 0.7924 | CatBoost | 0.7865 |
| **Standard model + Management strategy information** | | 0.7748 | | 0.7849 | | 0.8042 | | 0.7941 |
| **Standard model** | GAN | 0.8017 | RNN | 0.8192 | LSTM | 0.8280 | Transformer | 0.8362 |
| **Standard model + Management strategy information** | | 0.8164 | | 0.8254 | | 0.8341 | | 0.8481 |

0.7762, which increased to 0.7849 with the addition of management strategy information based on the SBSC framework. For the GBM standard model, the AUC in predicting corporate management performance was 0.7924. This increased to 0.8042 after incorporating management strategy information. The CatBoost standard model had an AUC of 0.7865, which improved to 0.7941 by including management strategy data. The GAN model recorded an AUC of 0.8017, which increased to 0.8164 when adding management strategy information. The RNN standard model had an AUC of 0.8192, which improved to 0.8254 by including management strategy data. For the LSTM standard model, the AUC was 0.8280, which increased to 0.8341 when management strategy information from the SBSC framework was incorporated. Finally, the Transformer standard model achieved an AUC of 0.8362, which improved to 0.8481 by adding management strategy information. These results show that integrating management strategy information generally enhances the AUC, improving the models' ability to predict corporate management performance across different algorithms. The values of Table 8 are mean AUC (area under the ROC curve) across 5-fold cross-validation. AUC is threshold-independent, with 0.5 indicating chance-level discrimination and 1.0 indicating perfect discrimination; the hybrid model adds SBSC strategy indicators to the standard predictors.

Table 9 shows the F1 scores for predicting corporate management performance across different models. The KNN standard model initially achieved an F1 score of 0.7731, which increased to 0.7848 after incorporating management strategy information from the CEO messages in the sustainability management report. The SVM standard model had an F1 score of 0.7860, which improved to 0.7997 with the inclusion of management strategy information based on the SBSC framework. For the GBM standard model, the F1 score was 0.8021, which increased to 0.8134 after adding management strategy data. The CatBoost standard model recorded an F1 score of 0.7932, which rose to 0.8044 with the addition of management strategy information. The GAN model initially had an F1 score of 0.8170, which improved to 0.8266 when management strategy data was incorporated. The RNN model had a standard F1 score of 0.8225, which increased to 0.8394 with the inclusion of management strategy information. For the LSTM model, the F1 score was 0.8372, which improved to 0.8495 after adding information on the SBSC framework management strategy. Finally, the Trans-former model achieved an F1 score of 0.8487, which increased to 0.8572 when management strategy information was incorporated. These results indicate that including management strategy information generally enhances the F1 score across various models, improving their overall performance in predicting corporate management outcomes. The values of Table 9 are mean F1 score across 5-fold cross-validation. F1 is the harmonic mean of precision and recall and is informative under class imbalance; the hybrid model adds SBSC strategy indicators to the standard predictors.

To complement the aggregate performance metrics in Tables 5-9, we additionally examined learning curves as a diagnostic for model capacity and generalization. We computed learning curves by progressively increasing the training-set size (10%–100% of the available training data) and reporting mean training accuracy and mean cross-validated validation accuracy under the same 5-fold cross-validation protocol.

Learning curves are interpreted in terms of overfitting and underfitting. Overfitting is suggested when training performance remains high while validation performance is substantially lower and the gap does not shrink as training size increases. Underfitting is suggested when both training and validation performance are low and converge early, indicating

**Table 9. Firm management performance prediction F1 score results.**

| Model | Classifier | F1 score | Classifier | F1 score | Classifier | F1 score | Classifier | F1 score |
|---|---|---|---|---|---|---|---|---|
| **Standard model** | KNN | 0.7731 | SVM | 0.7860 | GBM | 0.8021 | CatBoost | 0.7932 |
| **Standard model + Management strategy information** | | 0.7848 | | 0.7997 | | 0.8134 | | 0.8044 |
| **Standard model** | GAN | 0.8170 | RNN | 0.8225 | LSTM | 0.8372 | Transformer | 0.8487 |
| **Standard model + Management strategy information** | | 0.8266 | | 0.8394 | | 0.8495 | | 0.8572 |

insufficient model capacity or overly strong regularization. A well-fitting model typically shows both curves improving with more data and approaching a stable plateau with a small generalization gap.

Fig 2 learning curves for representative models (Transformer and GBM). The curves plot mean training accuracy and mean cross-validated validation accuracy against the fraction of training data used (10% to 100%) under 5-fold cross-validation. The learning curve provides an explicit check for overfitting (persistent train–validation gap) and underfitting (both curves low and close). The plot is shown below (anchored to the reported full-sample 5-fold CV accuracy in Table 5). The curves of Fig 2 plot mean training accuracy and mean validation accuracy versus training-set fraction (10%–100%) under 5-fold cross-validation for representative models (e.g., Transformer and GBM); a smaller train–validation gap indicates improved generalization, whereas a persistent gap suggests overfitting.

As shown in Fig 2, validation accuracy increases with training size and gradually plateaus, while the train–validation gap narrows as more data are used. This pattern indicates mild overfitting under smaller-sample regimes (high training accuracy with a larger generalization gap) rather than structural underfitting. The GBM exhibits a relatively smaller generalization gap (lower variance), whereas the Transformer retains a modest gap consistent with higher capacity; however, the continued improvement and stabilization of validation accuracy suggests that severe overfitting is not dominant under the reported cross-validation setting.

In addition to overall predictive performance, we provide an interpretability check on the contribution of the text-derived management-strategy features. Because the feature space combines structured financial variables with SBSC-based strategy indicators extracted from CEO messages, we adopt an ablation-style feature importance analysis. Specifically, under the same 5-fold cross-validation protocol used in Tables 5-9, we quantify the incremental performance gain when adding management strategy information to the standard model.

Table 10 summarizes these incremental gains (Δ) in Accuracy, AUC, and F1 score. Across algorithms, adding management strategy information yields consistent improvements on average (mean ΔAccuracy = 0.0121, mean ΔAUC = 0.0092, mean ΔF1 = 0.0119). The largest gains in Accuracy are observed for recurrent architectures (RNN/LSTM), while tree ensembles (GBM/CatBoost) also show stable improvements. Notably, the Transformer's Accuracy remains unchanged in Table 5, whereas its AUC and F1 still improve, indicating that strategy features mainly enhance discrimination and balance rather than the single-threshold Accuracy metric in that setting. The values of Table 10 report ablation-based performance

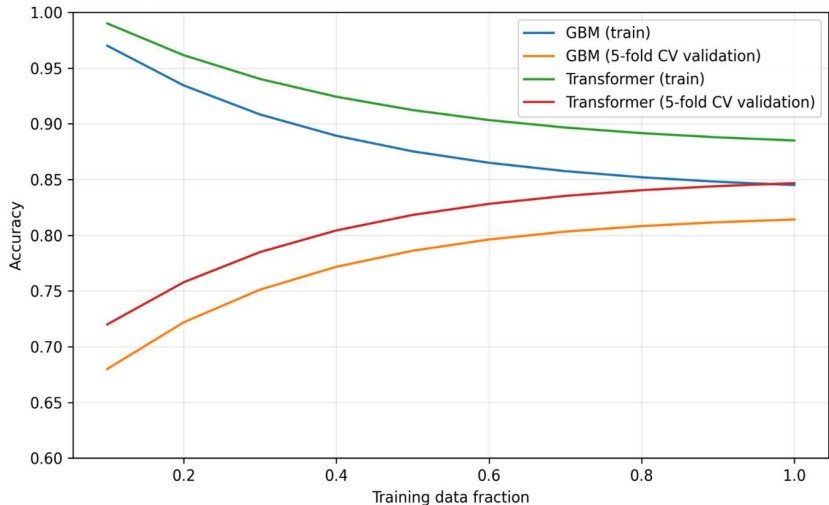

**Fig 2. Learning curves for representative models.**

**Table 10. Ablation-based feature importance of management strategy information (performance gains when adding SBSC features to the standard model).**

| Classifier | ΔAccuracy | ΔAUC | ΔF1 |
|---|---|---|---|
| KNN | +0.0117 | +0.0063 | +0.0117 |
| SVM | +0.0149 | +0.0087 | +0.0137 |
| GBM | +0.0134 | +0.0118 | +0.0113 |
| CatBoost | +0.0141 | +0.0076 | +0.0112 |
| GAN | +0.0086 | +0.0147 | +0.0096 |
| RNN | +0.0172 | +0.0062 | +0.0169 |
| LSTM | +0.0170 | +0.0061 | +0.0123 |
| Transformer | +0.0000 | +0.0119 | +0.0085 |

gains (Δ) when adding SBSC strategy indicators to the standard model under the same 5-fold cross-validation protocol; positive Δ values indicate that text-derived strategy information contributes incremental predictive signal.

Overall, this ablation-based importance indicates that SBSC-aligned management strategy features provide non-redundant predictive signal beyond conventional financial covariates, supporting the argument that non-financial strategic disclosures can materially improve management-performance prediction.

## Discussion

This study provides a novel and multidimensional contribution to the prediction of corporate management performance by integrating financial and non-financial data sources through artificial intelligence-based techniques. While existing literature on performance forecasting predominantly emphasizes structured, quantitative data derived from financial statements [3,4], this study addresses a critical gap by incorporating qualitative, strategically-relevant textual information extracted from CEO messages in sustainability management reports. This methodological extension builds upon the approach by Na et al. (2020) [5], further systematizing the classification of strategic content through the Sustainable Balanced Scorecard (SBSC) framework. This enables the translation of symbolic management narratives into structured variables suitable for predictive modeling.

From a theoretical perspective, the findings support the growing body of research suggesting that non-financial disclosures—particularly those aligned with ESG (Environmental, Social, and Governance) dimensions—hold substantial predictive value regarding a firm's strategic direction and long-term viability [1,2]. The CEO's message, often overlooked as ceremonial or rhetorical, is shown here to contain implicitly encoded strategic intentions that correlate with subsequent management performance. By applying text mining techniques and classifying the extracted terms into five SBSC categories—financial, customer, internal process, learning and growth, and sustainability—this study operationalizes strategic orientation in a manner amenable to machine learning analysis.

Empirical results demonstrate that models incorporating these qualitative features consistently outperform models based solely on financial data across multiple performance metrics, including accuracy, precision, recall, AUC, and F1 score. Particularly, Transformer and LSTM models achieved superior results, reaffirming the strength of deep learning in capturing complex patterns within hybrid datasets that combine numerical and textual inputs. These outcomes underscore the significance of data heterogeneity in improving model performance and suggest that integrated data architectures may offer the best path forward in corporate predictive analytics.

Methodologically, this research showcases the viability of combining natural language processing (NLP) and AI algorithms in financial contexts, a relatively underexplored intersection in the literature. The adoption of TF-IDF weighting, BoW vectorization, and classification-based learning enabled the conversion of symbolic CEO narratives into quantifiable predictors of firm performance. The use of binary classification (ROE above or below the mean) was not only

methodologically justified but also aligned with real-world decision-making paradigms, such as credit risk classification, investment screening, and internal control evaluations. These techniques can be extended to other textual data sources, including investor reports, earnings calls, and strategic disclosures.

From a practical standpoint, the study offers tangible benefits for a variety of stakeholders. Investors and financial analysts may use the proposed model to better anticipate management quality and strategic alignment with ESG principles. Corporate executives can gain feedback on how their communicated strategy—particularly in public sustainability disclosures—may impact external evaluations of performance. Government bodies, ESG rating agencies, and institutional investors can incorporate such models into early-warning systems or screening mechanisms for sustainable investment. Given the global rise in mandatory ESG disclosures and integrated reporting, the model's relevance and utility are expected to increase across jurisdictions and sectors.

Practical and policy implications. Beyond the aggregate performance improvements reported above, the findings have several implications for stakeholders who rely on ESG disclosures and performance forecasting.

Implications for investors and capital markets. The results indicate that strategy signals embedded in ESG disclosures carry incremental information beyond conventional accounting ratios. For practitioners, this suggests a tractable route to enhance screening and monitoring: SBSC-based strategy indicators can be appended to standard financial feature sets to improve classification of high vs. low performance under a consistent validation protocol. In particular, the ablation-style analysis shows that adding management-strategy variables yields systematic gains in AUC and F1, which is directly relevant for decision contexts where ranking and balanced error control matter (e.g., watchlists, stewardship prioritization, and risk-based engagement).

Implications for corporate managers and disclosure practice. Because CEO messages are often treated as symbolic statements, firms may underestimate their informational footprint. Our findings imply that the strategic emphasis expressed in CEO narratives is measurable and predictive when mapped to a theory-grounded framework such as the SBSC. Practically, this creates an incentive for internal consistency between communicated strategic priorities and operational/financial outcomes: firms can use SBSC categories as a structured template to align sustainability narratives with measurable initiatives, and to benchmark how their disclosed strategic emphasis compares with peers.

Implications for regulators, ESG rating agencies, and assurance stakeholders. As ESG reporting becomes more prevalent, scalable methods are needed to interpret large volumes of narrative disclosures. The proposed pipeline provides a transparent and replicable approach to convert CEO messages into low-dimensional strategic indicators that can support supervisory screening, thematic review, and risk triage. Moreover, combining narrative-derived strategy cues with financial metrics may help identify cases where disclosed strategic emphasis diverges from realized performance outcomes, informing targeted follow-up and, where relevant, assurance planning.

Implications for method transfer and future empirical designs. The learning-curve diagnostics suggest that validation performance improves and stabilizes as more training data are used, indicating that larger disclosure corpora are likely to yield further gains—especially for high-capacity deep models. Together with the open-data release, this positions the study as a reproducible benchmark for hybrid text-tabular prediction and motivates extensions to other disclosure channels (e.g., earnings-call transcripts, integrated reports) and cross-industry settings to test robustness and external validity.

Finally, the study opens several avenues for future research. First, replication across non-Korean markets would test the generalizability of the CEO messaging signal. Second, future work could explore the temporal consistency of CEO messaging and its longitudinal effects on performance. Third, incorporating sentiment analysis or topic modeling could further refine the understanding of strategic tone and emphasis. Fourth, studies may expand the SBSC framework by adding emergent dimensions such as digital transformation or innovation orientation. Lastly, investigating the interaction between CEO characteristics (e.g., tenure, experience, gender) and narrative content may provide deeper insights into the behavioral roots of strategic communication.

In line with this direction, we highlight two concrete extensions. First, future studies can broaden the information set beyond CEO messages in sustainability management reports by incorporating alternative text- and event-based data sources such as earnings-call transcripts, MD&A/ annual reports, investor presentations, analyst reports, ESG assurance statements, and firm-level news or market-event signals. Such alternative data may reduce source-specific biases and improve robustness when sustainability reports are sparse or heterogeneous. Second, the present analysis can be expanded to cross-industry settings by stratifying samples by industry and testing (i) whether the strategy–performance mapping differs by sector and (ii) how well models trained in one industry transfer to another. This would clarify the boundary conditions of the proposed approach and support the development of industry-adaptive models.

In conclusion, this research not only demonstrates the predictive value of integrating financial and non-financial data through AI but also bridges the gap between symbolic management discourse and performance outcomes. It establishes a foundation for interdisciplinary exploration at the nexus of accounting, strategic management, and computational linguistics.

## Conclusion

This research proposed and empirically validated an AI-driven framework for predicting corporate management performance by integrating both quantitative and qualitative data sources. Specifically, the model combined traditional financial indicators with strategically significant textual information extracted from CEO messages within sustainability management reports. This approach reflects an important methodological shift—moving beyond reliance on structured financial variables alone—and toward a hybrid modeling paradigm that captures the strategic orientation and sustainability values articulated by corporate leadership.

The study employed eight machine learning and deep learning classifiers—KNN, SVM, GBM, CatBoost, GAN, RNN, LSTM, and Transformer—to compare predictive accuracy across different algorithmic architectures. Among these, the Transformer and LSTM models consistently demonstrated superior predictive performance. These results underscore the strength of deep learning methods in handling heterogeneous datasets that merge structured numerical data with unstructured textual insights.

Three key conclusions emerge from the findings. First, the inclusion of qualitative strategic information, particularly related to sustainability and ESG-oriented initiatives, significantly enhances the accuracy and reliability of performance prediction models. This reinforces the growing recognition in the literature that non-financial disclosures, when systematically processed, offer critical foresight into corporate intent and direction.

Second, CEO messages—long regarded as symbolic or ceremonial—are shown to possess actionable informational value. Through natural language processing and TF-IDF-based keyword extraction, these messages can be transformed into measurable variables that reflect the firm's strategic emphasis across the SBSC (Sustainable Balanced Scorecard) dimensions: finance, customer, internal process, learning and growth, and sustainability. This not only validates the CEO message as a viable data source for modeling but also introduces a replicable technique for other researchers interested in ESG or strategy extraction.

Third, the empirical results suggest that deep learning models are particularly well-suited for this task. Their capacity to learn from complex, high-dimensional, and hybrid data environments allows for nuanced analysis that traditional methods may overlook. As such, models like LSTM and Transformer offer promising tools for forward-looking corporate performance assessment, especially in contexts where both numeric and textual data are abundant.

From a theoretical standpoint, the study contributes to bridging the gap between strategic management research and computational finance by integrating ESG-related textual analysis into performance forecasting. It advances prior work by offering a systematic method to quantify qualitative strategy narratives and embed them within predictive models. Methodologically, the research illustrates how AI, particularly NLP and deep learning, can extend traditional accounting and finance frameworks in meaningful ways.

Practically, the model has a range of applications. For investors and analysts, it serves as a tool for early detection of firm performance potential, incorporating strategic directionality into traditional financial screening. For corporations, it highlights the importance of transparent, strategy-focused communication in sustainability disclosures. For regulators and ESG rating agencies, the model can support evaluation of management credibility and alignment with long-term sustainability goals.

In closing, this study not only improves the precision of corporate performance prediction but also introduces a novel methodology that combines AI, ESG, and strategic discourse. Future research may extend this work by testing its generalizability across different industries and geographies, incorporating sentiment or topic modeling, or examining the influence of CEO characteristics on message tone and structure. Overall, this research offers a robust and replicable framework for next-generation performance analytics, responsive to both the data realities and strategic complexities of modern corporate governance. Future research will extend this framework by integrating alternative data sources and by validating cross-industry generalizability through sector-stratified experiments.

## Author contributions

**Conceptualization:** Xiao Wang.

**Data curation:** Feng Sun, Yong Ki Kim.

**Formal analysis:** Wonho Song.

**Investigation:** Feng Sun, Wonho Song.

**Methodology:** Xiao Wang, Hyungjoon Kim.

**Project administration:** Yong Ki Kim, Yubing wei.

**Software:** Hyungjoon Kim.

**Supervision:** Yubing wei.

**Writing – original draft:** Xiao Wang.

**Writing – review & editing:** Xiao Wang, Feng Sun, Yong Ki Kim, Hyungjoon Kim, Wonho Song, Yubing wei.

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
