## [Decision Letter · Decision Letter 0]

16 Dec 2024

PONE-D-24-38231Predicting Corporate Management Performance Using AI: Incorporating CEO Strategy Insights from Sustainable Management ReportsPLOS ONE

Dear Dr. wei,

Thank you for submitting your manuscript to PLOS ONE. After careful consideration, we feel that it has merit but does not fully meet PLOS ONE’s publication criteria as it currently stands. Therefore, we invite you to submit a revised version of the manuscript that addresses the points raised during the review process.

If applicable, we recommend that you deposit your laboratory protocols in protocols.io to enhance the reproducibility of your results. Protocols.io assigns your protocol its own identifier (DOI) so that it can be cited independently in the future. For instructions see: https://journals.plos.org/plosone/s/submission-guidelines#loc-laboratory-protocols. Additionally, PLOS ONE offers an option for publishing peer-reviewed Lab Protocol articles, which describe protocols hosted on protocols.io. Read more information on sharing protocols at . Additionally, PLOS ONE offers an option for publishing peer-reviewed Lab Protocol articles, which describe protocols hosted on protocols.io. Read more information on sharing protocols at https://plos.org/protocols?utm_medium=editorial-email&utm_source=authorletters&utm_campaign=protocols..

We look forward to receiving your revised manuscript.

Kind regards,

Burak Erkayman

Academic Editor

PLOS ONE

Reviewers' comments:

Reviewer's Responses to Questions

**Comments to the Author**

1. Is the manuscript technically sound, and do the data support the conclusions?

Reviewer #1: Yes

Reviewer #2: Yes

Reviewer #3: Partly

2. Has the statistical analysis been performed appropriately and rigorously? 

Reviewer #1: Yes

Reviewer #2: Yes

Reviewer #3: No

3. Have the authors made all data underlying the findings in their manuscript fully available?

Reviewer #1: Yes

Reviewer #2: Yes

Reviewer #3: No

4. Is the manuscript presented in an intelligible fashion and written in standard English?

Reviewer #1: Yes

Reviewer #2: Yes

Reviewer #3: No

5. Review Comments to the Author

Reviewer #1: 1- Abstract—it would be a good idea to include some accuracy values in the abstract.

2- In line 123, you mentioned you have selected words that have a TF-IDF value of 15 or higher. You should explain how you decided that value. In other words, why that particular value is meaningful for your study?

3- Your discussion on what hyperparameters you tuned is not very clear. I would suggest to expand that discussion.

4- In the conclusion section, you should highlight the limitations of your work and discuss how future studies can address those.

Reviewer #2: Dear Authors, I enjoyed reading your manuscript ,

Your study shows a clear and consistent improvement in the performance of machine learning and deep learning models when management strategy information (derived from the CEO’s messages in sustainability reports) is incorporated alongside financial data. This finding is significant as it highlights the added value of non-financial information in predicting corporate management performance. Furthermore, the results of all the evaluation metrics are consistent (accuracy, precision, recall, AUC, and F1 score), showing that integrating management strategy data leads to improvements in the predictive performance of most models. The results for deep learning models, particularly the Transformer, LSTM, and RNN, are quite interesting, they outperform traditional machine learning models, highlighting their suitability for handling complex, time-series data in corporate financial analysis. Plus, the use of the SBSC framework to incorporate management strategy information is quite innovative and novel, demonstrating the integration of qualitative data into quantitative predictive models.

I have just a few recommendations, most of them are how the sentences should be re-structured, the science itself is sound.

My suggestions are included in the attached file to the editors.

Reviewer #3: I have uploaded a file to the system for you. I suggest you revise the introduction, method, findings and conclusion sections of your study's abstract.

There is too much repetition in the text. Your abstract section does not express your work and the study you conducted on CEO views very well.

Using tables in some places may be better for the readability of the study.

There is no discussion section in your study. I see that you did not emphasize enough what the findings you obtained will bring to us and the difference from similar studies in the literature.

I suggest you redesign your study more carefully.

Please review the file.

Do not forget to discuss the findings you obtained more clearly and express the method more clearly and explicitly.

Your conclusion section may be more organized.

6. PLOS authors have the option to publish the peer review history of their article (what does this mean?). If published, this will include your full peer review and any attached files.). If published, this will include your full peer review and any attached files.

.

Reviewer #1: **Yes:**Md AmiruzzamanMd Amiruzzaman

Reviewer #2: No

Reviewer #3: No

While revising your submission, please upload your figure files to the Preflight Analysis and Conversion Engine (PACE) digital diagnostic tool, https://pacev2.apexcovantage.com/. PACE helps ensure that figures meet PLOS requirements. To use PACE, you must first register as a user. Registration is free. Then, login and navigate to the UPLOAD tab, where you will find detailed instructions on how to use the tool. If you encounter any issues or have any questions when using PACE, please email PLOS at . PACE helps ensure that figures meet PLOS requirements. To use PACE, you must first register as a user. Registration is free. Then, login and navigate to the UPLOAD tab, where you will find detailed instructions on how to use the tool. If you encounter any issues or have any questions when using PACE, please email PLOS at figures@plos.org. Please note that Supporting Information files do not need this step.. Please note that Supporting Information files do not need this step.

---

## [Author Response · Author response to Decision Letter 1]

15 Sep 2025

We sincerely thank the editor and reviewers for their valuable comments and suggestions.

We have carefully revised the manuscript based on all the feedback provided.

A detailed point-by-point response is included in the uploaded “Response to Reviewers” document.

---

## [Decision Letter · Decision Letter 1]

30 Dec 2025

PONE-D-24-38231R1Predicting Corporate Management Performance Using AI: Incorporating CEO Strategy Insights from Sustainable Management ReportsPLOS One

Dear Dr. wei,

Thank you for submitting your manuscript to PLOS ONE. After careful consideration, we feel that it has merit but does not fully meet PLOS ONE’s publication criteria as it currently stands. Therefore, we invite you to submit a revised version of the manuscript that addresses the points raised during the review process.

If applicable, we recommend that you deposit your laboratory protocols in protocols.io to enhance the reproducibility of your results. Protocols.io assigns your protocol its own identifier (DOI) so that it can be cited independently in the future. For instructions see: https://journals.plos.org/plosone/s/submission-guidelines#loc-laboratory-protocols. Additionally, PLOS ONE offers an option for publishing peer-reviewed Lab Protocol articles, which describe protocols hosted on protocols.io. Read more information on sharing protocols at . Additionally, PLOS ONE offers an option for publishing peer-reviewed Lab Protocol articles, which describe protocols hosted on protocols.io. Read more information on sharing protocols at https://plos.org/protocols?utm_medium=editorial-email&utm_source=authorletters&utm_campaign=protocols..

We look forward to receiving your revised manuscript.

Kind regards,

Sajid Anwar, Ph.D

Academic Editor

PLOS One

Journal Requirements:

Reviewers' comments:

Reviewer's Responses to Questions

**Comments to the Author**

1. If the authors have adequately addressed your comments raised in a previous round of review and you feel that this manuscript is now acceptable for publication, you may indicate that here to bypass the “Comments to the Author” section, enter your conflict of interest statement in the “Confidential to Editor” section, and submit your "Accept" recommendation.

Reviewer #1: (No Response)

Reviewer #2: All comments have been addressed

2. Is the manuscript technically sound, and do the data support the conclusions?

Reviewer #1: Yes

Reviewer #2: Yes

3. Has the statistical analysis been performed appropriately and rigorously? 

Reviewer #1: Yes

Reviewer #2: Yes

4. Have the authors made all data underlying the findings in their manuscript fully available?

Reviewer #1: Yes

Reviewer #2: Yes

5. Is the manuscript presented in an intelligible fashion and written in standard English?

Reviewer #1: Yes

Reviewer #2: Yes

6. Review Comments to the Author

Reviewer #1: This paper is interesting and presents a timely study. Overall, I have noticed several good points from this study. For example, the motivation for this study is interesting and well-formed. I liked the comprehensive comparisons of different models.

However, some sections require more attention and clear discussion to enhance the paper's quality.

-Line 125: You mentioned a threshold value of 1.5 for the TF-IDF. You should discuss how you came up with the value of the threshold.

- Also, a learning curve would be a good addition to your results section. Also, you should discuss the learning curve in terms of overfitting and underfitting.

I did not see any discussion on feature importance. Please consider adding some discussion on that.

A clear motivation for the GAN should be discussed; perhaps, you could refer to similar work.

The method section lacks justification for the different analysis tools.

I think the discussion section could be strengthened by adding more implications.

Reviewer #2: Well done authors, the manuscript looks much better now. The study shows reports a clear and well-documented improvement in the performance of ML and DNN models when management strategy info (derived from CEO's messages in sustainability reports ) is included alongside the financial data. It highlights the value of non-financial info in predicting management performances. I hope and look forward to seeing future work from your group on this topic which would explore alternative data sources and perhaps also test models across different industries .

7. PLOS authors have the option to publish the peer review history of their article (what does this mean?). If published, this will include your full peer review and any attached files.). If published, this will include your full peer review and any attached files.

.

Reviewer #1: **Yes:**Md AmiruzzamanMd Amiruzzaman

Reviewer #2: **Yes:**Dr Leon Gerard D'CruzDr Leon Gerard D'Cruz

---

## [Author Response · Author response to Decision Letter 2]

22 Jan 2026

We sincerely thank the Editor and the Reviewers for their valuable and constructive comments.

We have carefully reviewed all comments from Reviewer 1 and Reviewer 2, and revised the manuscript accordingly.

Detailed, point-by-point responses to each comment, along with corresponding changes in the revised manuscript, are provided in the attached documents entitled “Response to Reviewer 1” and “Response to Reviewer 2.”

We hope that the revisions adequately address the reviewers’ concerns, and we would be pleased to further revise the manuscript if additional clarification or modifications are required.

We would be grateful for any further suggestions or requests for clarification.

---

## [Decision Letter · Decision Letter 2]

3 Feb 2026

PONE-D-24-38231R2Predicting Corporate Management Performance Using AI: Incorporating CEO Strategy Insights from Sustainable Management ReportsPLOS One

Dear Dr. wei,

Thank you for submitting your manuscript to PLOS ONE. After careful consideration, we feel that it has merit but does not fully meet PLOS ONE’s publication criteria as it currently stands. Therefore, we invite you to submit a revised version of the manuscript that addresses the points raised during the review process.

If applicable, we recommend that you deposit your laboratory protocols in protocols.io to enhance the reproducibility of your results. Protocols.io assigns your protocol its own identifier (DOI) so that it can be cited independently in the future. For instructions see: https://journals.plos.org/plosone/s/submission-guidelines#loc-laboratory-protocols. Additionally, PLOS ONE offers an option for publishing peer-reviewed Lab Protocol articles, which describe protocols hosted on protocols.io. Read more information on sharing protocols at . Additionally, PLOS ONE offers an option for publishing peer-reviewed Lab Protocol articles, which describe protocols hosted on protocols.io. Read more information on sharing protocols at https://plos.org/protocols?utm_medium=editorial-email&utm_source=authorletters&utm_campaign=protocols..

We look forward to receiving your revised manuscript.

Kind regards,

Sajid Anwar, Ph.D

Academic Editor

PLOS One

Journal Requirements:

Reviewers' comments:

Reviewer's Responses to Questions

**Comments to the Author**

1. If the authors have adequately addressed your comments raised in a previous round of review and you feel that this manuscript is now acceptable for publication, you may indicate that here to bypass the “Comments to the Author” section, enter your conflict of interest statement in the “Confidential to Editor” section, and submit your "Accept" recommendation.

Reviewer #1: All comments have been addressed

2. Is the manuscript technically sound, and do the data support the conclusions?

Reviewer #1: Yes

3. Has the statistical analysis been performed appropriately and rigorously? 

Reviewer #1: Yes

4. Have the authors made all data underlying the findings in their manuscript fully available?

Reviewer #1: Yes

5. Is the manuscript presented in an intelligible fashion and written in standard English?

Reviewer #1: Yes

6. Review Comments to the Author

Reviewer #1: You should consider adding a brief summary of key performance results to the abstract so readers can quickly grasp the main contributions and outcomes of the study.

The introduction would benefit from a more explicit justification of the need for this study. At present, the motivation is implied, but strengthening it by grounding the discussion in existing and recent research would be helpful. In particular, clearly identifying a gap in the literature and explaining how your work fills or extends that gap would improve the positioning of the paper.

The discussion on dataset adequacy is currently somewhat limited and would benefit from further elaboration. Please provide a more thorough justification of why the dataset is appropriate for this study. Additionally, you should more clearly justify the choice of the approaches and methods used, including why they are suitable compared to alternative options.

Finally, consider expanding the figure and table captions so that they are more self-contained. Well-developed captions can help readers better understand the purpose and interpretation of each figure and table without needing to refer extensively to the main text.

7. PLOS authors have the option to publish the peer review history of their article (what does this mean?). If published, this will include your full peer review and any attached files.). If published, this will include your full peer review and any attached files.

.

Reviewer #1: **Yes:**Md AmiruzzamanMd Amiruzzaman

---

## [Author Response · Author response to Decision Letter 3]

26 Feb 2026

Dear Editor,

We would like to thank you and the reviewers for the valuable comments and suggestions, which have helped us improve the quality of our manuscript.

Please find our detailed, point-by-point responses to the reviewers' comments in the attached "Response to Reviewers 1" file.

---

## [Decision Letter · Decision Letter 3]

29 Mar 2026

Predicting Corporate Management Performance Using AI: Incorporating CEO Strategy Insights from Sustainable Management Reports

PONE-D-24-38231R3

Dear Dr. wei,

We’re pleased to inform you that your manuscript has been judged scientifically suitable for publication and will be formally accepted for publication once it meets all outstanding technical requirements.

An invoice will be generated when your article is formally accepted. Please note, if your institution has a publishing partnership with PLOS and your article meets the relevant criteria, all or part of your publication costs will be covered. Please make sure your user information is up-to-date by logging into Editorial Manager at Editorial Manager® and clicking the ‘Update My Information' link at the top of the page. For questions related to billing, please contact  and clicking the ‘Update My Information' link at the top of the page. For questions related to billing, please contact billing support..

Kind regards,

Sajid Anwar, Ph.D

Academic Editor

PLOS One

Additional Editor Comments (optional):

Reviewers' comments:

Reviewer's Responses to Questions

**Comments to the Author**

1. If the authors have adequately addressed your comments raised in a previous round of review and you feel that this manuscript is now acceptable for publication, you may indicate that here to bypass the “Comments to the Author” section, enter your conflict of interest statement in the “Confidential to Editor” section, and submit your "Accept" recommendation.

Reviewer #1: All comments have been addressed

2. Is the manuscript technically sound, and do the data support the conclusions?

Reviewer #1: Yes

3. Has the statistical analysis been performed appropriately and rigorously? 

Reviewer #1: Yes

4. Have the authors made all data underlying the findings in their manuscript fully available?

Reviewer #1: Yes

5. Is the manuscript presented in an intelligible fashion and written in standard English?

Reviewer #1: Yes

6. Review Comments to the Author

Reviewer #1: Thank you for making the suggested changes. The paper reads well now. The manuscript is now well-organized and flows clearly.

7. PLOS authors have the option to publish the peer review history of their article (what does this mean?). If published, this will include your full peer review and any attached files.). If published, this will include your full peer review and any attached files.

.

Reviewer #1: **Yes:**Md AmiruzzamanMd Amiruzzaman

---

## [Editor Report · Acceptance letter]

PONE-D-24-38231R3

PLOS One

Dear Dr. wei,

I'm pleased to inform you that your manuscript has been deemed suitable for publication in PLOS One. Congratulations! Your manuscript is now being handed over to our production team.

Kind regards,

on behalf of

Dr. Sajid Anwar

Academic Editor

PLOS One